# Encapsulation and Characterization of Nanoemulsions Based on an Anti-oxidative Polymeric Amphiphile for Topical Apigenin Delivery

**DOI:** 10.3390/polym13071016

**Published:** 2021-03-25

**Authors:** Tzung-Han Chou, Daniel Setiyo Nugroho, Jia-Yaw Chang, Yu-Shen Cheng, Chia-Hua Liang, Ming-Jay Deng

**Affiliations:** 1Department of Chemical and Materials Engineering, National Yunlin University of Science and Technology, Yunlin 64022, Taiwan; danielsetiyanugraha@gmail.com (D.S.N.); yscheng@gemail.yuntech.edu.tw (Y.-S.C.); 2Department of Chemical Engineering, National Taiwan University of Science and Technology, Taipei 10607, Taiwan; jychang@mail.ntust.edu.tw; 3Department of Cosmetic Science, Chia Nan University of Pharmacy and Science, Tainan 717, Taiwan; tinna_ling@mail.cnu.edu.tw; 4Department of Applied Chemistry, Providence University, 200 Taiwan Boulevard, Sec. 7, Taichung 43301, Taiwan; dengmj1020@pu.edu.tw

**Keywords:** Apigenin, encapsulation efficiency, nanoemulsions, TPGS, antioxidant ability, rheological characteristics

## Abstract

Apigenin (Apig) is used as a model drug due to its many beneficial bio-activities and therapeutic potentials. Nevertheless, its poor water solubility and low storage stability have limited its application feasibility on the pharmaceutical field. To address this issue, this study developed nanoemulsions (NEs) using an anti-oxidative polymeric amphiphile, d-α-tocopheryl polyethylene glycol 1000 succinate (TPGS), hydrogenated soy lecithin (HL), black soldier fly larvae (BSFL) oil, and avocado (AV) oil through pre-homogenization and ultrasonication method. Addition of TPGS (weight ratios 100 and 50% as compared to HL) into NEs effectively reduced particle size and phase transition region area of NEs with pure HL. Incorporation of Apig into NEs made particle size increase and provided a disorder effect on intraparticle molecular packing. Nevertheless, the encapsulation efficiency of NEs for Apig approached to about 99%. The chemical stability of Apig was significantly improved and its antioxidant ability was elevated by incorporation with BSFL oil and AV oil NEs, especially for NEs with single TPGS. NEs with single TPGS also exhibited the best Apig skin deposition. For future application of topical Apig delivery, NEs-gel was formed by the addition of hyaluronic acid (HA) into NEs. Their rheological characteristics were dominated by the surfactant ratios of HL to TPGS.

## 1. Introduction

Apigenin (Apig) is one of the natural dietary flavonoid compounds originating from the Apiaceae family [1], and it can be found in fruits, vegetables, and herbs [2]. Apig has been proven to play a potent role in biological activities such as effective wound healing [3], treatment for skin inflammation, and repairing UV-B-induced DNA damage in human skin [4]. However, Apig is classified as a BCS (Biopharmaceutical Classification System) II drug with low water solubility that hinders its bioavailability [5]. Thus, it becomes a big challenge to effectively penetrate Apig into skin layer.

To overcome the problem of drug solubility, different carriers such as ethosomes [6], nanoparticle [7], liposomes [8], and O/W emulsions [9] have been introduced to encapsulate Apig. Nevertheless, employing liposomes and ethosomes on drugs may induce unexpected side effects in humans [10,11], and their protectional performances on the chemical stability of Apig are still not very clear. Among the delivery systems mentioned above, nanoemulsions (NEs) are a suitable vehicle for Apig because they exhibit high potential to improve the water solubility and bioavailability of several oil-soluble phytochemicals including quercetin [12], catechin [13], β-carotene [14], and emodin [15]. Furthermore, nanoemulsification is regarded as a promising strategy for promoting drug skin penetration without additional chemical enhances, where the nanoemulsion compositions (i.e., oils, surfactants, and co-surfactants) themselves can act as penetration enhancers on transdermal drug absorption [16]. NEs have been proven to increase drug deposition rate in skin [17] and to prolong the drug activity itself [18]. Therefore, the utilization of nanoemulsions is a good candidate for application of topical Apig delivery.

Usually, oils are considered as a structural core of NEs. Black soldier fly larvae (BSFL) oil and avocado (AV) oil are used herein to prepare NEs because both of them are able to form stable NEs in our previous works [19,20]. BSFL oil is a naturally occurring animal oil from black solder fly (*Hermetia illucens*) larvae. It possesses an antimicrobial activity against growth of Gram-negative bacteria, Gram-positive bacteria, and filamentous fungi [21], and it contains a high percentage of linoleic acid and oleic acid that can increase skin permeation [22]. AV oil is rich in omega fatty acids and shows many more pharmacological characteristics such as hypercholesterolemia management, cardiometabolic risk management, hypertension management, diabetes management, hepatoprotective effect, antimicrobial activity. Well refined AV oil may show a good skin penetration ability because it owns mono-unsaturated oleic acid [22]. To further enhance the thermodynamic stability and drug solubility of colloidal dispersions, binary surfactants are generally used [23,24]. D-α-tocopheryl polyethylene glycol 1000 succinate (TPGS) and hydrogenated soy lecithin (HL) utilized as emulsifiers in NEs with Apig. TPGS is an amphiphilic derivative of natural vitamin E through esterification of vitamin E succinate, showing anti-oxidative characteristic. TPGS has been reported in various applications like, for example, as an anticancer agent [25], P-gp inhibitor [26], or additive in fabricating nanocarrier [27]. Hydrogenated soy lecithin (HL) shows negligible cytotoxicity effect and has been widely applied on preparation of liposomes [28] and as an emollient enhancer for lotion applications [29]. It can be expected that incorporation of TPGS and HL into NEs can provide antioxidant capacity and safety benefits on this Apig carrier.

Many studies have pointed that polymeric gelling agents are widely applied in the biomedical field [30], and they have pointed out the enhancement of drug percutaneous penetration in vitro and in vivo [31,32]. Hyaluronic acid (HA) one of polysaccharides known as a biological origin material, has excellent biocompatibility, non-immunogenicity, and biodegradability, which make it suitable for medical applications [33]. Besides, it has been confirmed that HA could not only control drug release but also improve transdermal absorption performance [34]. Herein, NEs gels were developed by incorporated NEs with HA for future topical Apig application. The rheological characteristics of gels would affect the drug release behavior and interaction time of drug with skin, and those played a key role in determination of transdermal drug absorption [35]. Therefore, rheological characteristics of these HA-NEs gels were measured in this work.

In this study, BSFL oil- and AV oil- NEs with various formulation compositions for encapsulation of Apig were manufactured by pre-homogenization combined with an ultrasonication way, and their physicochemical characteristics and drug encapsulation were determined. The effects of surfactant ratios of HL to TPGS on morphological observation, size, zeta potential, phase transition behavior, encapsulation performance, and chemical stability of BSFL oil and AV oil NEs loaded with Apig were investigated comprehensively. The antioxidant capacity and skin deposition amount of Apig of these NEs varied with different mixed surfactant ratios were also examined. Additionally, BSFL oil and AV oil NEs-HA gels were fabricated and their rheological properties were investigated deeply. This study hopes to provide useful information on the development of Apig-loaded carriers for topical delivery.

## 2. Materials and Methods

### 2.1. Materials

BSFL oil was provided by Wormax Inc, Taiwan. AV oil was purchased from Storino’s Quality Product( Glendale, Arizona, USA). TPGS was purchased from Eastman( South Wales, UK). HL was supplied from Merck KGaA(Darmstadt, Germany). Uranyl acetate was purchase from Electron Microscopy Science(Hatfield, Britain). Apig (5,7-dihydroxy-2-(4-hydroxyphenyl)-4H-1-benzopyran-4-one) was purchased from Tokyo Chemical Company CO., LTD.(Tokyo, Japan). Hyaluronic Acid was purchased from Bloomage Freda Biopharma co., LTD., Taiwan. Methanol was purchased from Avantor Performance Materials Inc. (Radnor Township, Pennsylvania, USA). Ethanol was purchase from Echo Chemical CO., LTD., Taiwan. Sodium chloride was purchased from Showa Chemical CO., LTD., (Tokyo, Japan). Artificial skin membrane was purchased from Merck Millipore Ltd., Co Cork, Ireland. Ultra-pure water produced by a Mill-Q-plus purification instrument (Burlington, Massachusetts, USA) was used for all experiments.

### 2.2. Preparation of NEs and Apig Loaded NEs

The previous process [20] with some modification was adopted to prepare BSFL oil and AV oil NEs. Briefly, oils, surfactants and aqueous phase were pre-heating in 80 °C in 10 min then homogenized at 15,000 rpm for 2 min using D-500 homogenizer (Wiggen Hausser, Germany). Further, the emulsified dispersion was sonicated under a power of 80 W at 45 °C for 30 min with an ultra-sonicator (Misonix 3000,Farmingdale, New York, USA). 10 mL of blank NEs was prepared by oil content 1%, the emulsifier consisted of surfactant to oil ratio (SOR) 60%, and surfactant ratio of HL/TPGS at 0/100, 50/50, and 100/0. The Apig loaded NEs were prepared using the same procedure as described above with addition of 2 mg Apig in the oil phase followed with stirring using vortex mixer. Each formulation repeated at least three times.

### 2.3. Transmission Electron Microscopy (TEM) Characteristics

Morphological characteristics of NEs were analyzed using a transmission electron microscope (HITACHI H-7500, Tokyo, Japan) at 100 kV. Before each TEM experiment, 5 μL of NEs was spread on the copper grid coated with carbon for 10 min and then negatively stained with 2 μll of uranyl acetate solution (2.5%, *w*/*w*). The sample was kept for 3min at room temperature and subsequently remained in an electronic dried box before observation.

### 2.4. Droplet size, PdI, Zeta Potential Measurement, and Physical Stability Analysis

Average particle size (APS) and polydispersity index (PdI) were determined using Zeta Plus Analyzer (Brookhaven Instruments Corporation, Holtsville, NY, USA) and followed by zeta potential measurement. Fresh empty NEs and Apig-loaded NEs were pipetted 1 mL and appropriately diluted in pure water to reach the test criteria. Hydrodynamic particle size and polydispersity index were obtained by 10 min of automatic cycle calculation and the zeta potential was measured by phase analysis light scattering combined with electrophoresis technique. At least three independent measurements were performed for each sample and reported as the mean±standard deviation. Besides, the storage stable days of NEs at 24.5 °C were determined by average particle size (APS) < 500 nm, polydispersity (PdI) <0.4, or no phase separation through optical observation.

### 2.5. Thermal Phase Change Behavior of NEs

Thermal phase transition behavior of NEs were determined using differential scanning calorimetry (DSC) (DSC 1 instrument with the STARe system, Mettler Toledo Inc., Greifensee, Switzerland), controlled by a Huber TC45 immersion cooler. Approximately 20 µL samples were placed in aluminum sealed pans, while 20 µL water-sealed pans used as a reference. The pans were placed in the DSC chamber with an initial temperature of 25°C. The thermal analysis was carried out over a temperature range of −10°C to 80°C with 5°C/min heating-cooling rate in N_2_ gas. All samples were analyzed with three heating-cooling cycles and repeated in triplicate (n=3).

### 2.6. Encapsulation Characteristics of NEs for Apig

Apig amounts loaded into NEs were assayed by HPLC instrument. The encapsulation efficiency (EE) and loading efficiency (LE) of NEs were calculated through the following equations:EE (%) = (total drug amount − free drug amount)/total drug amount × 100%(1)
LE (%) = (total drug amount − free drug amount)/total oil amount × 100%(2)

In brief, HPLC apparatus equipped with a model L-2200 autosampler, L-2130 pump, L-2455 diode array detector, and a temperature controller. The column of Luna^®^ 5µm C18(2) 100Å was used and pure methanol as the mobile phase flow at a flow rate of 1.0mL/min. The temperature set to be 35 °C and the detection wave number ranged from 200 nm to 700 nm was used and specific Apig wavenumber at 267 nm was marked. Every sample was filtered through a PVDF 0,22 µm (Milipore Millex-GV) filter. The free Apig content was determined by measuring the amount of untrapped Apig in the aqueous phase after ultrafiltration centrifugation of NEs at 15000rpm and 4 °C for 2 hr with a 5K MWCO column (Vivaspin, GE healthcare) placed on an Eppendorf Centrifuge 5424 R.

### 2.7. Chemical Stability Analysis

The chemical stabilities of pure Apig dispersion and Apig-loaded NEs were investigated by using a UV-Vis spectrophotometer (SPECTROstar^®^ Nano BMG LABTECH GmdH, Ortenberg, Germany). The study was done by comparing the sample absorbance with methanolic Apig solution (5% *v*/*v* methanol). Absorbance wavelengths at 267 nm and 336 nm were measured in the 0, 1, 2, 3, 4, 6, 12, 18, 24, 30, 36 h.

### 2.8. Antioxidant Activity Assay

The antioxidant activities of pure Apig dispersion, NEs encapsulated with Apig, were determined by using free radical-screening activities of DPPH assay, and the method was based on previous report [36] with some modifications. 0.024 g/L of DPPH solution was prepared and then reacted with samples including Apig and the Apig-loaded NEs. 0.1 mL of the samples were added to 1.9 mL of the DPPH solutions and kept in the dark place. Sampling was done for each 10 min in the period of 40 min and their absorbance values were measured at 517 nm with a UV-Vis spectrophotometer. Thus, the percentage inhibition on DPPH free radical activity (% *inhibition*) was calculated using equation below:(3)$$\%\;inhibition=\;\left(\frac{ABS_{control}-ABS_{sample}}{ABS_{control}}\right)\times 100$$

Where, *ABS_control_* is the absorption value of control DPPH and *ABS_sample_* is the absorption value of reacted DPPH.

### 2.9. Drug Penetration Study

A skin drug penetration of Apig-loaded NEs was determined using a Franz diffusion cell through artificial skin Strat-M^®^ Membrane 25 mm discs. The receptor compartment was filled with medium contained 40% ethanol and 0.9% NaCl. Volume of 5 mL medium was added ad maintained at 37 °C by circulating bath with a small magnetic stirrer was placed in the receptor chamber and set the speed at 700 rpm. The artificial skin was use directly without any pretreatment and fitted to the diffusion cell. 1 mL of prepared NEs then added to the top compartment. Using HPLC at 1, 2, 3, 6, 10, 16, 24-h, 0.5 mL of receptor medium was withdrawn and analyzed. To further analyze the Apig deposition on the artificial skin, the membrane was removed from the Franz cell and cleaned using a paper wipe. Next, the membrane was cut into small pieces and put into a bottle with methanol and ultrasonicated for 20 min. Before HPLC analysis, the sample was centrifuge at 2000 rpm for 15 min and filtered through 0.22 µm filter.

### 2.10. Preparation of NEs-HA Gels

Before incorporation of HA with NEs, blank NEs with SOR = 30 and 60% and surfactant ratios of HL to TPGS = 0/100, 50/50, and 100/0 were prepared as described early. Three and six percent *w*/*w* of hyaluronic acid in blank NEs were scaled and added to the NE samples. Then, a vortex mixer was used to strongly disperse HA into the NEs. To ensure complete dispersion of HA, the rotary-mixer equipment was introduced for every sample for 60 min. Finally, the vacuum instrument was used to remove the bubble from the NEs-HA gels for avoiding evade void effect on rheology measurement.

### 2.11. Rheology Study

Rheological investigation was performed by HR-2 Discovery Hybrid Rheometer (DHR) TA instrument, equipped with a plate geometry (40mm with temperature-controlled base). To investigate the rheological behavior, different hyaluronic concentration (3% and 6%, *w*/*w*) was added to NEs as mentioned above. Before the rheological analysis, the stability tests of NEs (Appendix A) were carried through a modified method as reported elsewhere [37,38]. Then, linear viscoelastic (LVE) region was measured by amplitude sweep and followed with frequency sweep test. Several presumptions were applied such as 25 °C observation temperature, 0.1–100 rad/s for frequency sweep radian range, and 15 sampling point per decades. After HA incorporated into NEs, the storage modulus (G’) and loss modulus (G”) as a function of strain rate and frequency were measured.

### 2.12. Statistical Analysis

In this work, all experiments were repeated at least three times and the data were represented as the mean value ± standard deviation. The results were statistically analyzed by ANOVA test and *p* values <0.05 were considered as significant.

## 3. Results and Discussion

### 3.1. Morphology Assay

Morphological appearance of nanosized colloids as well as their size was studied under a microscopy analysis. TEM images of Apig loaded AV-NEs and BSFL-NEs are shown in Figure 1. Spherical droplets were observed in the studied sample. The comparations of average particle size determined from the TEM images and DLS measurement showed no significant difference. The pictures also confirmed the possible existence of micelle in the dispersion indicated with the presence of particles with diameter under 10 nm. The droplets ware distributed moderately homogeneous throughout the formulations with slight aggregation observed in several dispersion systems. These observations also indicated that the Apig-loaded nanoemulsion was fabricated successfully.

In addition, the result suggested that using TPGS as a standalone surfactant would exhibit an aggregation phenomenon where the small particle diffusing to the larger particle. In colloidal systems, this situation can be caused by the solubility of a droplet increases notably as its radius becomes smaller [39]. An interesting finding was shown in the mixed surfactant system, where aggregation phenomena was eliminated and each particle has a better dispersive characteristic. This behavior may attribute to the increasing absolute zeta potential value by the addition of HL and decreasing the number of PEG which decreases the dispersion solubility in water [40].

### 3.2. DLS Characterization and PHYSCIAL Stability of Nanoemulsions Encapsulated with Apig

The average particle size, polydispersity, and zeta potential of BSFL oil and AV oil NEs are listed in Table 1. The average particle size of NEs added with TPGS (weight ratios 100 and 50% as compared to HL) were quite smaller than those prepared with only HL. Similarly, other works found that addition of TPGS to the colloidal system could decrease the particle size and increase its stability [36,41]. However, the BSFL oil NEs were found to be slightly smaller in size than the AV oil NEs, which was probably affected by the higher content of saturated fatty acid in BSFL oil. It can be seen that the particle sizes of AV oil NEs or BSFL NEs with pure TPGS systems increase triplicate from before to after Apig encapsulation. However, an increased degree on the size of NEs decreased with increasing HL amount, while the PdI shifted inconsistency and the ZP value increased for all formulation. This finding suggested that the Apig successfully was encapsulated into the AV oil NEs or BSFL NEs. The infusion of Apig into the particles cost the multiplication in the particle size and increasing the zeta potential. In addition, it might be that the Apig was not completely soluble in the oil phase or that the Apig solubility in oil was over its saturation point.

Additionally, Table 1 shows that the number of stable days for BSFL oil and AV oil NEs with the addition of TPGS is much higher than those without TPGS, no matter before or after Apig encapsulation. This might be because the PEG side chains of TPGS can provide a steric effect between NEs and a downsizing effect of TPGS on NEs may increase their surface charge density (possibly enhancing the electrostatic repulsion between NEs). Such similar TPGS effects on other colloidal systems were also reported elsewhere [36,42]. However, one could find that the storage stability of NEs incorporated with Apig was lower than those without Apig. This indicated that Apig might give a strong disturbed effect on the intra-NEs molecular packing, resulting in the low storage stability of NEs.

### 3.3. Endothermic Phase Change Behavior

Influences of TPGS content and Apig on the endothermic phase change behavior of AV oil and BSFL oil NEs were determined by DSC. Figure 2 shows endothermic DSC curves of AV oil- and BSFL oil- NEs with different surfactant ratios of HL to TPGS at SOR 60% and oil 1%. Single phase transition region was found in the temperature range of 56–68 °C for the AV oil NEs (marked by P2 region), whereas two phase transition regions of −4 to 37 °C (marked by P1 region) and 56–68 °C (P2) were obtained for BSFL oil NEs. According to previous reports [16,20], P1 and P2 are associated with BSFL oil and pure HL dispersions, respectively. At fixed oil content and SOR, one can find that the peak size of P1 or P2 region decreases with increasing TPGS ratio. Also, the enthalpy change of P1 or P2 region decreases with increasing TPGS content (Table 2). This effect may account for the fact that TPGS would provide a steric effect on the molecular packing within the dispersion core. A similar effect of TPGS on molecular arrange of micelle systems is also reported elsewhere [41]. While there was no observed transition for AV oil and BSFL oil NEs, an other report considered a completely soluble TPGS in a mixture of water and oil [43]. Moreover, after incorporations of Apig to the NEs system, the transition areas for all regions were consistently decrease by comparison with the DSC results reported previously [14,15]. This may cause by an increase in intermolecular interactions between Apig and TPGS/HL oils in a compacted structure, resulting in the occurrence of disordered molecular packing. In the other hand, the transition shape became broader after Apig encapsulation which might induce an increase of particle size and less homogeneous size distribution. These results can also be supported by the other literature [44], where a long-range order crystal structure of lycopene disappeared after it was encapsulated into NEs.


### 3.4. Encapsulation Performance of Apig

The encapsulation efficiency (EE) and loading efficiency (LE) of BSFL oil and AV oil NEs with different surfactant ratios of HL/TPGS (0/100, 50/50, and 100/0) have been determined as listed in Table 1. One can find that the EE and LE of BSFL oil and AV oil NEs for Apig approached to about 99% and 1.99% for all formulations, respectively. It was confirmed that the Apig encapsulation was handled with optimally under nanoemulsion preparation. One can see that the free Apig was less than 1% which is in a row with the nature of low solubility of Apig in water. Besides, high energy preparation method would provide sufficient energy to generate high sear stress which could rupture the viscous droplets and facilitate the emulsifications [45]. Different oils and surfactant ratios would not affect the EE and LE, where it provided an opportunity to increase the drug loading in the system. Besides that, it could be suggested that the Apig was well-interacted with TPGS, HL, and oils, where one could not find any bulk phase separation or drug dissoluble peak in DSC curves.

### 3.5. Chemical Stability Analysis of Apig Loaded into Nanoemulsions

To determine whether AV oil NEs and BSFL oil NEs can protect the chemical activity of Apig, their impacts on chemical degradation during room temperature storage were monitored. The percentage of remaining Apig concentration during storage is measured with certain intervals of time as shown in Figure 3. As a pure Apig solution in 5% methanol, its concentration percentage quickly decayed to 40% at the first 12 h and remained constant until 36 h. The occurrence of chemical degradation activity of Apig was found, possibly resulting from Apig oxidation in the environment of abundant water and air.

On the other hand, incorporation of Apig into the NEs greatly enhanced its chemical stability up to >84% after 36 h in the TPGS mediated AV oil NEs. The Apig degradation then seemed faster at the initial stage and relatively remained constant after 6 h. BSFL oil NE using HL as standalone surfactant showed lower remaining chemical stability of Apig (approximately 73% compared as the initial concentration), while utilization of mixed surfactants (HL/TPGS) maintained at 76% for BSFL oil NEs. Similar effect of TPGS content on the protection ability of Apig was also found in AV oil NEs. TPGS has an α-tocopheryl structure which may provide an antioxidant activity [46]. The result suggested that incorporation of Apig into BSFL oil and AV oil NEs could avoid Apig quickly degradation. Besides, the degradations rate of Apig was related with the droplet size of both NEs by compared with the results of DLS. Such a similar finding was also reported on Apig encapsulation in ethyl oleate O/W emulsions [9]. It has been reported that larger droplets surface possibly induced higher diffusion of Apig across the interface, resulting in Apig degradation.

### 3.6. Antioxidant Ability Assay

The antioxidant activities of Apig, blank NEs and NEs encapsulated with Apig were determined by the DPPH radical sequestration method, as shown in Figure 4. The blank BSFL oil NEs presented an antioxidant activity ranged from 6.4 to 9.3%, while empty AV oil NEs have a lightly lower inhibition percentage ranged from 4.1 to 7.5%. With increasing TPGS content in blank NEs and Apig-lead NEs, their antioxidant activities were elevated. Such enhanced effect of TPGS on antioxidant activity of NEs could be supported by the finding in the Section 3.6. Furthermore, BSFL oil NEs with or without Apig showed slightly higher antioxidant activity than AV oil NEs. This may attribute to the fact that AV oil used herein was refined oil in which the value of radical scavenging activity was reduced. This inference was able to be supported by the other study [47], which found that refining process could decrease the antioxidant activity through eliminating minor constituents such as phenols, oryzanol, phytosterols and tocopherol.

### 3.7. Artificial Skin Deposition of NEs Encapsulated with Apig

The drug penetration ability of NEs in the skin generally was influenced by drug solubility nature and the formulation composition. Franz diffusion cell assay with artificial skin membrane was performed to evaluate the skin permeation/retention ability of Apig-loaded NEs. Because Apig content cannot be detected in the receptor medium, the only skin deposition of Apig-loaded BSFL oil NEs and AV oil NEs with different surfactant ratios of HL/TPGS is shown in Figure 5, whatever the formulation. It was shown that AV oil NEs performed a similar permeated drug percentage as BSFL oil NEs with a fixed surfactant. However, an increase in the TPGS content promoted Apig deposition percentage in the skin membrane for BSFL oil and AV oil NEs. This might be attributed to the downsize effect of NEs by addition of TPGS. In addition, an other study reported that the colloidal size played a key role in dermal drug delivery [48].

### 3.8. Rheology Behavior of Nanoemulsions in HA Gel

The spreadability of topical formulations is highly related to their rheological behavior. Figure 6 and Figure 7 illustrate the frequency-dependence of G’ and G” of BSFL oil and AV oil NEs with HA gels, respectively, under a constant oscillation strain of 1%. One can see that the rheological behavior of NEs-HA gels was significantly impacted by different surfactant ratios of HL to TPGS impact rather than HA concentration or SOR for both BSFL and AV oils NEs. As addition of pure TPGS with NEs in a fixed 6% of HA, increasing SOR from 30% to 60% will slightly decrease the crossover modulus point and increasing the G’ value, which may be related to a decrease in the surface tension. This similar phenomenon was reported elsewhere, in which an increase in dispersant concentration would lead to an increase on G’ value of the gel formulations [49].

NEs with pure HL in HA gels exhibited an elastic behavior, where G’ exceeded G’’ over the entire frequency range, indicating the formation of a gel-like substance. This behavior was probably caused by weak repulsive interaction between HA and surface of NEs-HL. However, NEs added with mixed surfactants of HL/TPGS = 50/50 in HA gels expressed G”> G’ over the entire frequency, reflecting their viscous nature. Their G’ values initially increased but had a sharp downturn at a certain high frequency, suggesting that there might be an occurrence of network rupture with loss of elasticity. This implied that HL/TPGS (50/50) cosurfactants would produce strong repulsive interaction between HA network and NEs, possibly inducing a phase separation. Additionally, NEs with pure TPGS in HA gels were viscoelastic samples, where the elastic (G’) and viscous (G”) moduli of the sample crossed at a certain frequency, implying a relaxation time. The magnitudes of both G’ and G” were increased; however, the moduli expressed a strong frequency dependence. This rheological characteristic was very different from that of NEs with the addition of HL in HA gels. Particularly, there is no plateau in G’ at long timescales, implying the samples were able to relax. Their rheology was indicative of a transient network formed by junction at a finite time scale. A similar phenomenon was also found in other colloidal systems [50].

## 4. Conclusions

In the present study, Apig-loaded NEs with BSFL and AV oils as well as mixed HL/TPGS surfactants were successfully prepared by using pre-homogenization and ultrasonication methods. Apig was efficiently encapsulated into NEs. The particle size and peak area of phase transition of BSFL oil and AV oil NEs with pure HL can be effectively reduced with increasing TPGS amount. The chemical stability and antioxidant ability of Apig was significantly improved by incorporation with TPGS-BSFL oil and AV oil NEs. Addition amount of TPGS not only impacted the skin permeation behavior of NEs for Apig but also dominated rheological behavior of NEs-HA gel. In summary, this study can provide very useful information on the development of topical delivery systems for Apig.

## Figures and Tables

**Figure 1 polymers-13-01016-f001:**
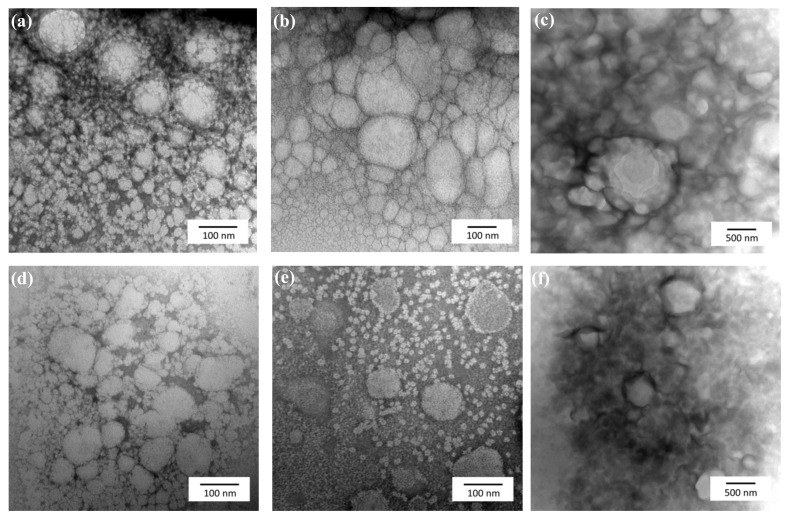
TEM images of Apig incorporated into BSFL oil NEs with SOR 60%, oil content 1%, and surfactant ratios of HL/TPGS = (**a**) 0/100, (**b**) 50/50, and (**c**) 100/0. TEM images of Apig incorporated into AV oil NEs with the same composition ratios as described above as surfactant ratios of HL/TPGS = (**d**) 0/100, (**e**) 50/50, (**f**) 100/0.

**Figure 2 polymers-13-01016-f002:**
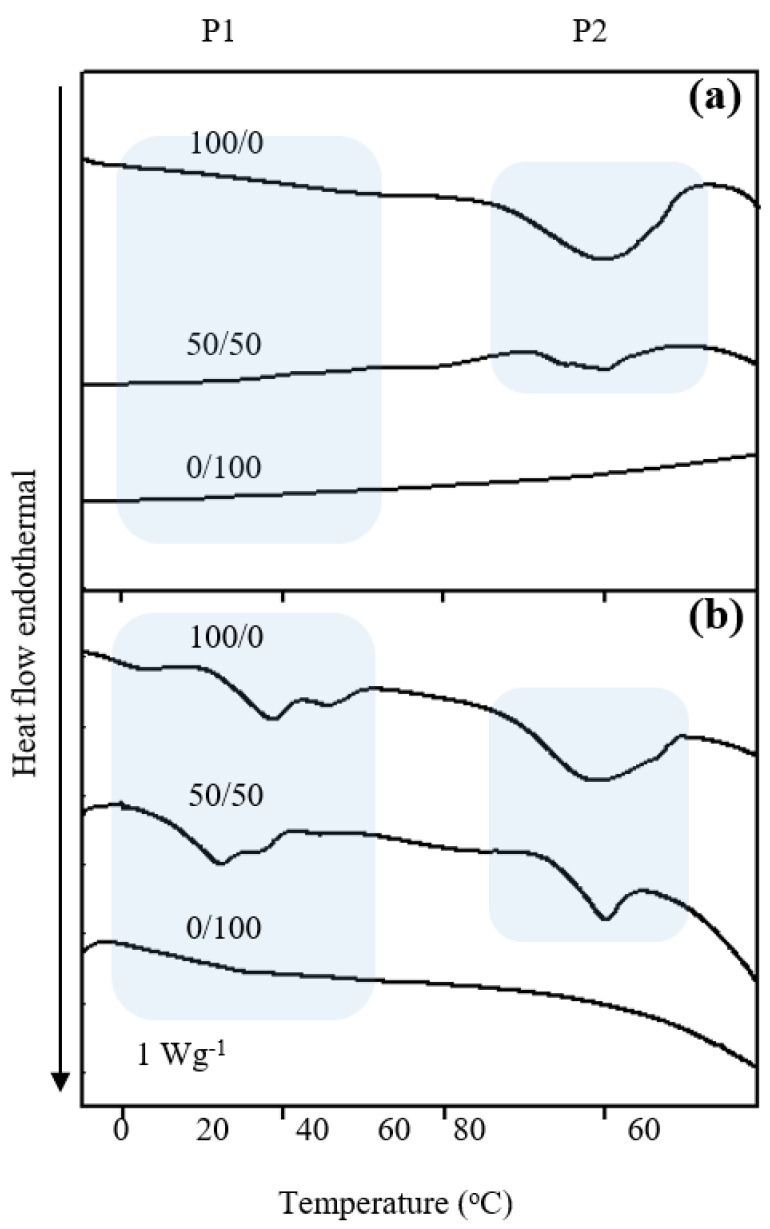
Effects of surfactant ratio of HL/TPGS on endothermal DSC diagrams of (**a**) AV oil NEs and (**b**) BSFL-NEs encapsulated with Apig at SOR 60% and oil 1%.

**Figure 3 polymers-13-01016-f003:**
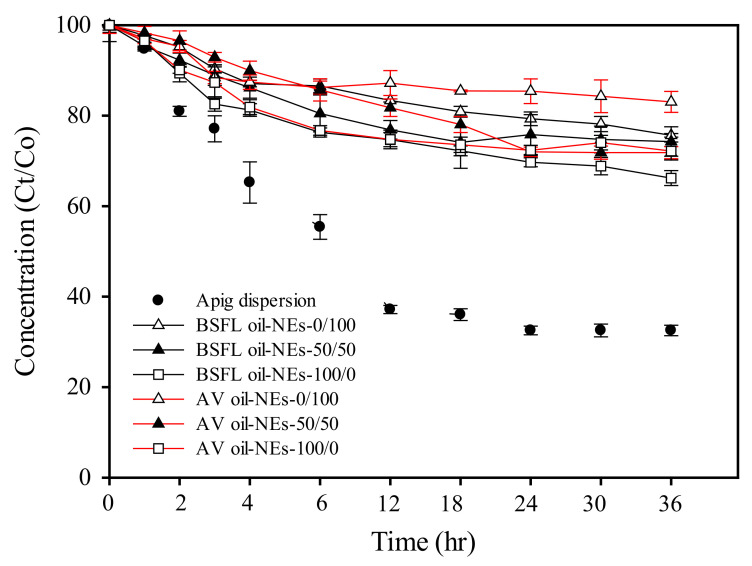
Chemical storage stability of Apig incorporated into BSFL oil NEs and AV oil NEs with SOR 60%, oil content 1%, and different surfactant ratios of HL to TPGS.

**Figure 4 polymers-13-01016-f004:**
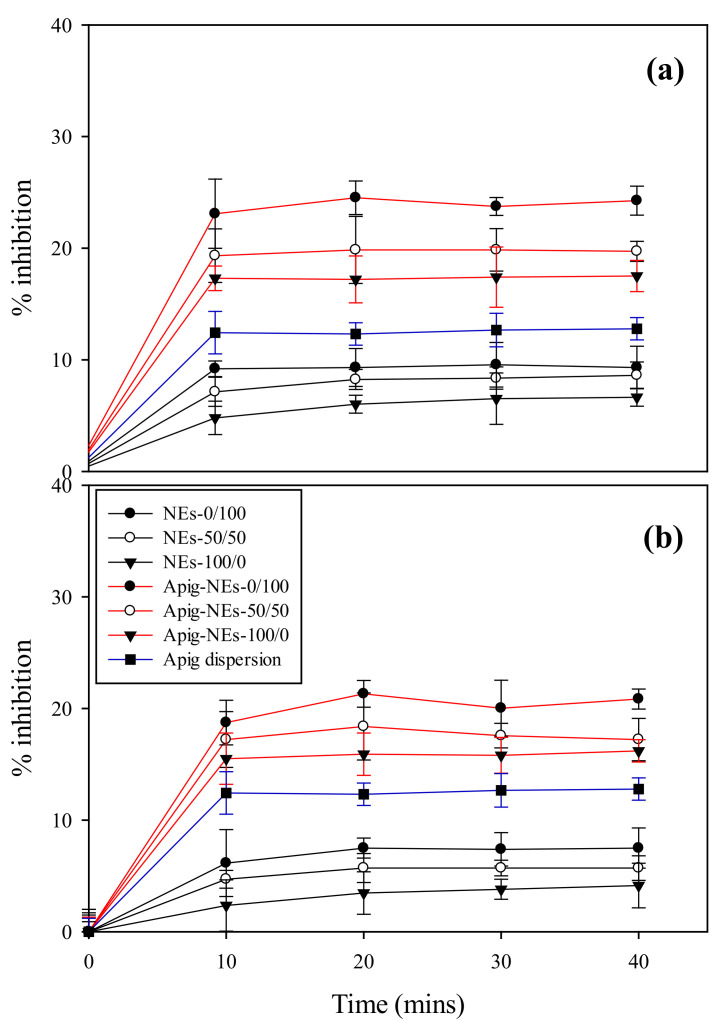
Antioxidant activities of Apig, blank NEs, and Apig loaded (**a**) BSFL oil NEs and (**b**) AV oil NEs with different surfactant ratios of HL/TPGS at SOR 60% and oil content 1%.

**Figure 5 polymers-13-01016-f005:**
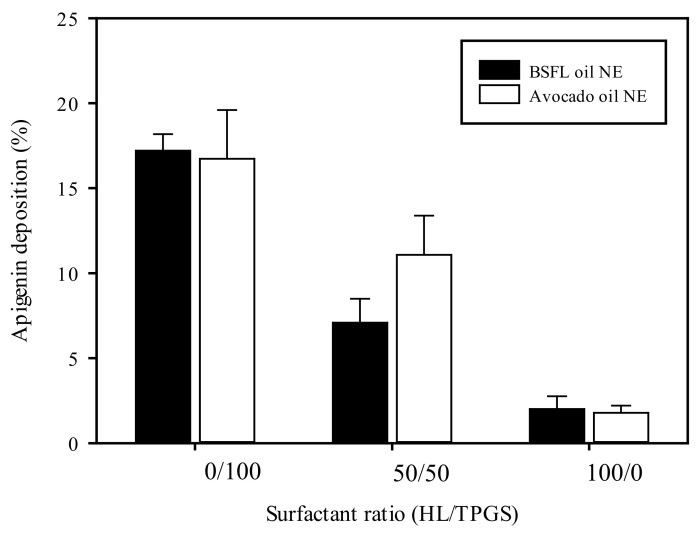
Artificial skin deposition of Apig loaded BSFL oil NEs and AV oil NEs with different surfactant ratios of HL/TPGS at SOR 60% and oil content 1%.

**Figure 6 polymers-13-01016-f006:**
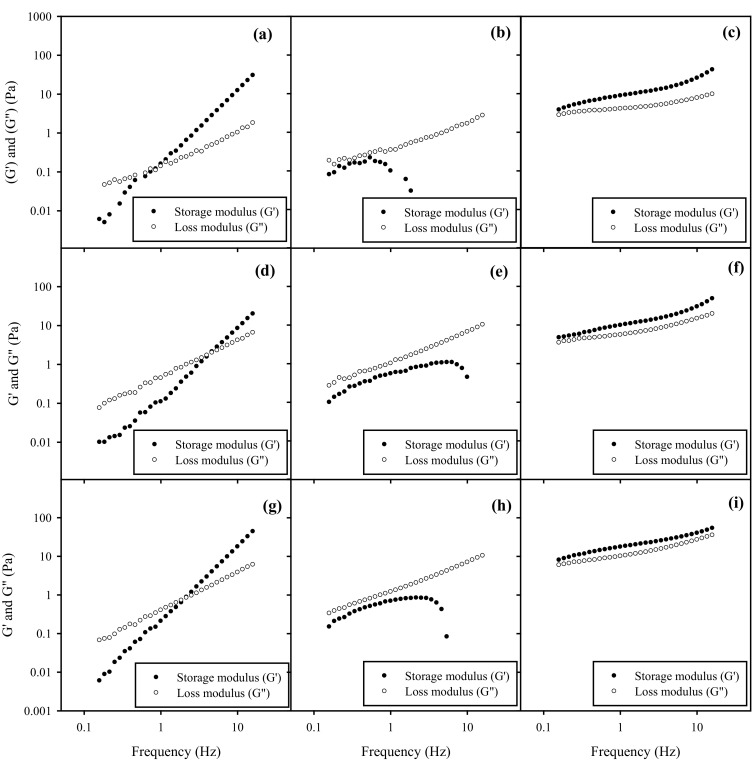
Frequency sweep on storage modulus ● (G’) and loss modulus ○ (G”) of BSFL oil NEs with SOR 30% and HL/TPGS ratios for (**a**) 0/100, (**b**) 50/50, (**c**) 100/0 in the 3% HA gel. G’ and G” of BSFL oil NEs with SOR 30% and HL/TPGS ratios for (**d**) 0/100, (**e**) 50/50, and (**f**) 100/0 in the 6% HA gel. G’ and G” of BSFL oil NEs with SOR 60% for (**g**) 0/100, (**h**) 50/50, and (**i**) 100/0 in the 6% HA gel.

**Figure 7 polymers-13-01016-f007:**
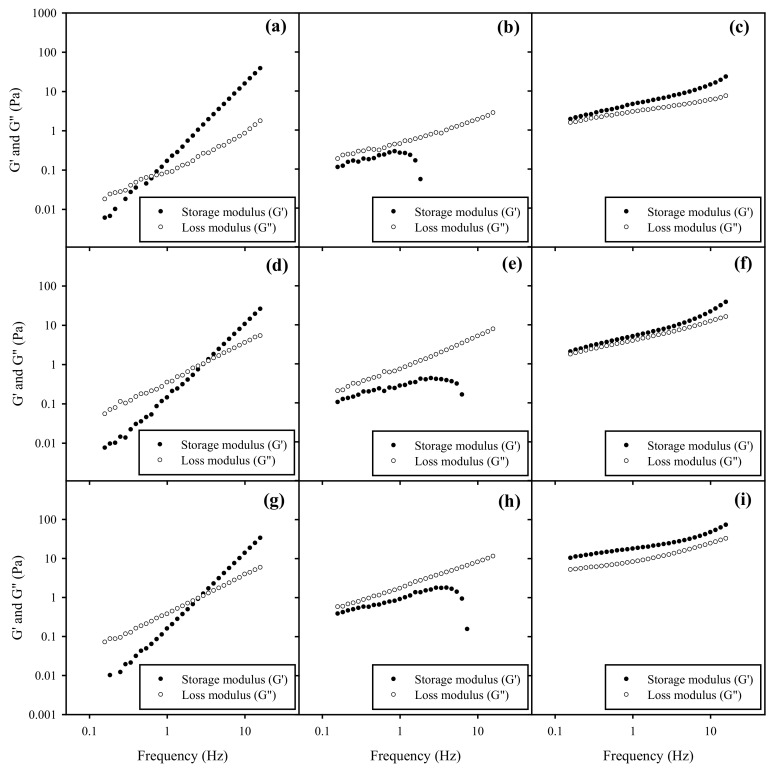
Frequency sweep on storage modulus ● (G’) and loss modulus ○ (G”) of AV oil NEs with SOR 30% and HL/TPGS ratios for (**a**) 0/100, (**b**) 50/50, (**c**) 100/0 in the 3% HA gel. G’ and G” of AV oil NEs with SOR 30% and HL/TPGS ratios for (**d**) 0/100, (**e**) 50/50, and (**f**) 100/0 in the 6% HA gel. G’ and G” of AV oil NEs with SOR 60% for (**g**) 0/100, (**h**) 50/50, and (**i**) 100/0 in the 6% HA gel.

**Table 1 polymers-13-01016-t001:** Encapsulation efficiency and loading efficiency of BSFL oil NEs and AV oil NEs with SOR 60%, oil content 1% and surfactant ratios of HL/TPGS for Apig with their average particle size (APS) nm, polydispersity (PdI), zeta potential (ZP) mV, and stable days (SD) by comparison before with after encapsulation.

Oil	HL/TPGS		Before Encapsulation	After Encapsulation	EE (%)	LE (%)
BSFL oil	0/100	APS: PdI: ZP : SD:	32.13 ± 0.83 0.19 ± 0.01 −22.2 ± 0.4 210 ± 7	135.5 ± 10.9 0.21 ± 0.02 −35.1 ± 5.9 45 ± 3	99.72 ± 0.01	1.994 ± 0.02
	50/50	APS: PdI: ZP : SD:	51.20 ± 3.36 0.35 ± 0.01 −41.9 ± 0.8 200 ± 7	122.1 ± 7.9 0.26 ± 0.04 −51.5 ± 6.4 30 ± 2	99.69 ± 0.01	1.993 ± 0.02
	100/0	APS: PdI: ZP: SD:	195.20 ± 0.80 0.16 ± 0.02 −43.0 ± 0.7 5 ± 1	230.4 ± 13.0 0.24 ± 0.01 −65.8 ± 3.6 3 ± 1	99.39 ± 0.01	1.987 ± 0.03
AV oil	0/100	APS: PdI: ZP : SD:	32.80 ± 3.54 0.19 ± 0.08 −13.1 ± 0,02 346 ± 10	120.1 ± 8.8 0.21 ± 0.03 −30.1 ± 1.7 56 ± 4	99.64 ± 0.05	1.99 ± 0.11
	50/50	APS: PdI: ZP : SD:	48.76 ± 9.75 0.33 ± 0.02 −41.2 ± 0.6 300 ± 7	118.39 ± 2,6 0.25± 0.03 −43.3 ± 4.1 40 ± 3	99.81 ± 0.01	1.99 ± 0.02
	100/0	APS: PdI: ZP : SD:	231.8 ± 4.2 0.20 ± 0.03 −50.1 ± 1.64 9 ± 1	234.7 ± 7.31 0.19 ± 0.01 −63.6 ± 2.2 5 ± 1	99.74 ± 0.09	1.99 ± 0.18

**Table 2 polymers-13-01016-t002:** Enthalpy changes (∆H), main transition temperature (∆Tm), and half-width transition temperature (∆T_1/2_) of BSFL oil and AV oil nanoemulsions encapsulated with Apig under SOR = 60% and different surfactant ratios of HL/TPGS on P1 and P2 phase transition regions obtained from endothermal DSC curves.

Oil	HL/TPGS	Tm^p1^	Tm^p2^	∆H^p1^ (mJ)	∆H^p2^ (mJ)	∆T_1/2_^p1^	∆T_1/2_^p2^
**BSFL oil**	0/100	14.65	-	0.95	-	4.83	-
	50/50	12.13	60.05	4.48	2.05	4.81	2.37
	100/0	18.84	60,41	5.77	8.82	3.39	5.75

**AV oil**	0/100		-		-		-
	50/50		59.91		2.58		5.19
	100/0		59.48		11.78		7.44

-: Data cannot be determined.

## Data Availability

The data presented herein are available on request from the corresponding author.

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
