# Peer review of "Encapsulation and Characterization of Nanoemulsions Based on an Anti-oxidative Polymeric Amphiphile for Topical Apigenin Delivery"

_polymers, 2021, doi:10.3390/polym13071016_

Round 1
Reviewer 1 Report
The paper “Apigenin encapsulation of nanoemulsions based on an anti-oxidative polymeric amphiphile and their rheological characteristics in hyaluronic acid hydrogel” by Chou et al. deals with the preparation and characterization of some oil-in-water emulsions stabilized by two surfactants, such as TPGS and HL at different weight ratios.
The study is interesting but rather basic as only one ratio between the two phases was investigated. In addition, the language is quite poor and needs extensive editing. Other corrections are:
- Line 23: please rephrase (weight ratios of 100 and 50% as compared to HL)
- Line 40: delete Shen2014. Please revise the references from the introduction section.
- Which is the total volume of a typical emulsion? This information must be added in the section 2.2
- Line 112: replace “in order to evaluate the TEM images” with “in order to evaluate the droplet size” or “in order to compare the droplet sizes obtained by TEM”
- Line 167: the abbreviation for apigenin must be provided in the introduction section and then used for the entire manuscript
- Line 259: which compound has changed the crystallinity?! Please complete
- Fig 3: the values for the x-axis are not visible. Idem for fig 5
- Concerning the rheological analysis, the authors must carry out some stability tests (viscosity vs shear rate and viscosity vs time at a constant shear rate). Moreover, they should clearly indicate if the viscosity of their emulsions allow the spreadability on the skin. More information here: https://doi.org/10.1016/j.ijpharm.2013.03.051; https://doi.org/10.1016/j.colsurfa.2014.01.026
- Which is the stability of the obtained nanoemulsions in the presence and in the absence of drug?!
Reviewer 2 Report
The manuscript “Apigenin encapsulation of nanoemulsions based on an antioxidative polymeric amphiphile and their rheological characteristics in hyaluronic acid hydrogel” deals with the encapsulation of Apigenin into nanoemulsion and the addition of a natural-based gel for topical delivery, with the aim of improving drug bioavailability. The technologies used for this scope were pre-homogenization and ultrasonication. Rheological analyses on the produced carriers were also performed, obtaining intriguing results. However, the manuscript requires some revisions and improvements.
In particular:
- The title of the manuscript is not clear. Please, semplify it.
- Introduction. The state of the art has to be widely enlarged to highlight the novelty of this work and the characteristics of nanoemulsions and hydrogels. For this purpose, see, for instance, the work of Baldino et al., A new tool to produce alginate-based aerogels for medical applications, by supercritical gel drying, Journal of Supercritical Fluids, 2019, 146, pp. 152-158; etc…
The scope of the work is not clear. Please, rewrite.
- Results and Discussion. SEM images of the nanocarriers should be added to better observe their morphology.
It is not clear the preparation method used to add HA in nanoemulsions. Please, describe in a dedicated paragraph.
- Some typing errors are present. Please, correct them.
- English can be improved.
Author Response
Please seen the attachment.

Round 2
Reviewer 1 Report
The paper can be published as it is.
Reviewer 2 Report
The authors performed the modifications proposed by the Reviewer and improved the manuscript.